# Advanced Maternal Age: A Scoping Review about the Psychological Impact on Mothers, Infants, and Their Relationship

**DOI:** 10.3390/bs14030147

**Published:** 2024-02-20

**Authors:** Monica Ahmad, Cristina Sechi, Laura Vismara

**Affiliations:** Department of Pedagogy, Psychology, Philosophy, University of Cagliari, 09123 Cagliari, Italy; monica.ahmad@unica.it (M.A.); cristina.sechi@unica.it (C.S.)

**Keywords:** advanced maternal age, pregnancy outcomes, perinatality, maternal mental health, protective factors

## Abstract

The mean age at childbirth in Europe has gradually increased, and it is now around 29 years of age. It has been shown that older maternal age is associated with problems of fertility; in fact, with increasing age, the chance of conceiving diminishes, and fetal and obstetric complications grow. Research has focused particularly on the biological risks associated with late pregnancy, both for the child and the woman. Less space has been dedicated to the potential psychological and relational benefits of motherhood at an advanced age. The aim of this review was to summarize the existing literature on this issue. Qualitative and quantitative studies were sourced from Pubmed, Science Direct, PsycINFO, and SciELO. The selected works highlight that advanced maternal age can be associated with some advantages for both mothers and their offspring in terms of physical healthcare, parenting styles, and child developmental outcomes. Specifically, the review suggests that older mothers have greater emotional maturity and feel more prepared for motherhood; also, advanced maternal age appears to exert a protective influence on children’s behavioral, social, and emotional functioning, compensating for the biological risks.

## 1. Introduction

In recent years, the age of first-time mothers has significantly increased. Among the possible reasons for this, many researchers propose that it may be caused by the greater participation of women in higher education and in employment and in their desire to reach financial stability; indeed, these motivations may lead women to postpone the moment in which they decide to become mothers [1,2,3]. In Europe, the average age of first-time mothers registered in 2021 was 29.7 years, a significant increase from previous years, as reported by Eurostat [4]. 

The increased maternal age carries with it problems of infertility since the ability to conceive spontaneously decreases as a woman’s age advances [5]. Research has shown that the optimal age range for successful conception and healthy childbirth is 20–34 years old. Fertility steeply declines over the age of 35 when conceiving a child becomes increasingly difficult [6]. 

In addition, older maternal age is also associated with higher rates of pregnancy complications and poorer obstetrical and neonatal outcomes. Among these risks, most studies indicate gestational diabetes, gestational hypertension, placental detachment, placenta previa, miscarriage, low birth weight, preterm or post-term birth, and postpartum hemorrhages [7,8,9,10,11,12,13,14,15,16,17]. Therefore, the scientific literature confirms the association between advanced maternal age and poor obstetric and infant outcomes. 

In addition, giving birth at an older age is linked to psychological conditions such as autism, bipolar disorder, and symptoms of depression, anxiety, and stress, along with impaired social functioning [18].

Despite the recognition of the critical importance of the early parent–child relationship quality for children’s socioemotional, cognitive, neurobiological, and health outcomes, there is currently limited exploration of the psychological impacts of the advanced maternal age on mothers, infants, and their relationships. 

The goals of this review are as follows:(a)To understand the perinatal psychological experience in advanced-age mothers;(b)To study the parenting behaviors of advanced-age mothers compared to younger mothers.

## 2. Materials and Methods

This research was conducted as a scoping review. A scoping review is useful to map the literature on evolving topics and identify gaps. They follow the guidelines for systematic reviews but are simplified to facilitate consultation with stakeholders [19].

### 2.1. Search Strategy

Pubmed, Science Direct, PsycINFO, and SciELO were searched from their inception to September 2023 using the following keywords: advanced maternal age, maternal age and offspring, pregnancy outcome, postponement of childbirth, and parenting. No time limits were set to select the studies since the main purpose of the current review was to explore all the available empirical and clinical data. Articles were included if they fulfilled the following PICOS (population, intervention or exposure, comparison, outcomes, study design) eligibility criteria.

### 2.2. Population

Pregnant women or those who had a child at ≥35. Additionally, women under the age of 35 are considered for the purpose of comparing pregnancy outcomes between those of advanced maternal age and younger women.

### 2.3. Intervention/Exposure

Studies focusing on the perinatal psychological experience and parenting behaviors in the target population were considered.

### 2.4. Comparison

The review aimed to compare advanced-age mothers’ experiences during perinatal and those of younger mothers, both in terms of psychological functioning and parenting quality.

### 2.5. Outcomes

We looked at the following outcomes: parenting behaviors, perinatal distress in advanced maternal-age women, and psychological symptomatology (e.g., self-reported depression, anxiety, stress, and perception of pregnancy risks).

### 2.6. Study Design

We included quantitative studies, qualitative studies, literature reviews, and metanalyses.

### 2.7. Selection Criteria

The search had the following inclusion criteria: (1) quantitative/qualitative studies; (2) literature reviews; (3) metanalyses with advanced maternal age as their primary or secondary goal; and (4) published in English. Case studies and editorials were excluded, as well as articles that did not consider psychological aspects during pregnancy and abstracts without the full text available.

The initial search yielded 184 articles. Following the exclusion of duplicates and papers lacking the full text, 72 potentially relevant studies were evaluated for eligibility. After excluding any out-of-scope manuscripts, 24 articles were included in the review. MA and LV authors independently screened the full texts to mitigate selection bias. A narrative synthesis was employed to analyze pertinent papers categorized under themes. The study selection process is visually represented in Figure 1, as depicted in the PRISMA flow chart.

### 2.8. Data Extraction

The characteristics of included studies are shown in Table 1 and Table 2.

The tables have six entries for each study as follows: title, authors, sample, research design, objective, and results. Table 1 presents studies examining the obstetric risks associated with pregnancy in advanced age, with a secondary emphasis on psychological variables. By contrast, Table 2 compiles studies where psychological outcomes of late pregnancy are the primary focus, examining their impact on women and/or their offspring.

## 3. Results

### 3.1. Countries

The studies in the review include participants from Denmark (1 study) [29], the UK (2 studies) [21,34], Australia (4 studies) [26,32,33,35], Japan (one study) [25], Canada (2 studies) [22,23], Norway (1 study) [20], Sweden (1 study) [28], Brazil (2 studies) [36,37], and Spain (1 study) [24].

### 3.2. Participants

All studies involved in this review considered women who were at least 18 years old during the perinatal period. As regards advanced maternal age, the included studies considered women from 35 years of age or older. 

### 3.3. Instruments

Concerning the administered methods, all studies adopted self-reports, and four of these [23,31,32,33] used semi-structured, face-to-face interviews. Most studies applied multiple measures. Depression was assessed using the Edinburgh Postnatal Depression Scale [38] and the Beck Depression Inventory Short Form (BDI/SF) [39]. Parenting stress was assessed with the Parenting Stress Index Short Form (PSI/SF) [40]. Regarding anxiety, the tools used were the State-Trait Anxiety Inventory (STAI) [41], the five-item ‘Anxiety concerning Health and Defects in the Child’ scale from the Baby Schema Questionnaire [42] and the Pregnancy-Related Anxiety Questionnaire [43].

Other quantitative measures included the following: the Maternal–Fetal Attachment Scale (MFAS) [44] to measure the construct of maternal–fetal attachment during pregnancy; the Personal Views Survey (PVS-III-R) [45] to assess psychological hardiness; the 24-Item Intimate Bonds Measure (IBM) [46] assesses the perceived quality of the partner relationship; the Family Adaptability Cohesion Evaluation Scale (FACES) [47] to evaluate the adaptability and cohesion dimensions in family interactions, the Mother–Child Relationship Evaluation (MCRE) [48] to assess maternal attitudes; the Multidimensional Scale of Perceived Social Support (MSPSS) [49], a self-report measure of subjectively assessed social support; the Perception of Pregnancy Risk Questionnaire (PPRQ) [50]; the SF-12 Health Status Survey to measure functional health and well-being [51]; the Multidimensional Health Locus of Control Questionnaire (MHLC) [52] to measure perceived internal control; the Prenatal Scoring Form; and the Cambridge Worry Scare (CWS) to measure women’s concerns or worries during pregnancy [53].

### 3.4. Overall Psychological Experience in Women in Advanced Maternal Age

As regards the first objective, which is the mothers’ psychological experience, it has been shown that pregnant women aged 35 or older perceive their age as a risk factor for their pregnancy. However, these concerns seem to emerge only in the presence of other risk factors like unfavorable screening test results, previous poor reproductive history, or anxiety [23,33]. Older women within a healthy context and without other risks tend to perceive their pregnancy as low-risk [23]. Another survey comparing the risk perception in pregnant women aged 35 years or older to that of younger women showed that women of advanced maternal age had a higher perception of pregnancy risk compared to younger women, both for themselves and for the fetus [22]. However, the survey shows no significant difference compared to younger women with respect to perinatal-specific anxiety, health status, perceived control, and knowledge of pregnancy risks [22].

Maternal awareness of risk is a key element that can also have a positive influence on their decisions about when to seek medical attention and when to take appropriate measures and actions that can significantly improve maternal and fetal health and development.

Some longitudinal studies showed that older maternal age is associated with decreased risks of child hospital admissions and accidental injuries in the first three years of life [25,34], with better levels of language development and fewer emotional difficulties compared to younger women’s children [34]. Moreover, children of older mothers show higher educational levels and higher cognitive abilities [30,54]. The improved social resources of advanced-aged mothers might, in part, explain such findings.

From the above results, it can be stated that older women show high coping abilities, which may mitigate the obstetrical risks [21,55,56]. 

Indeed, several studies seem to confirm that postponing motherhood may represent a psychological advantage to both mothers and their children. Older mothers feel more capable of taking care of their children and more competent [31,36,57]; they also feel fulfilled in their role and consider pregnancy as an experience permeated by perceptions and feelings of satisfaction and achievement [31,32,33,36]. These women consider themselves as more prepared, more patient, tolerant, resilient, balanced, and emotionally ready, as well as more secure and competent with their infant [23,26,31,32,36,37]. They also describe their partners as less controlling [26].

### 3.5. Psychological Distress

By contrast, results are mixed in relation to the presence of psychological distress. Some studies report lower symptoms of depression and anxiety during pregnancy, a decrease in postpartum depressive symptoms with advancing age, and no significant differences between older and younger women in pregnancy-related anxiety, knowledge of risk, perceived control, and health status [22,26,28,58]; others showed higher concerns during pregnancy [28,35], including more depressive symptoms during the prenatal period [35,59], higher levels of postpartum depression [20,27,59,60,61] and stress [27,60,61] and slightly higher scores of psychological distress [59]. As regards the perception of pregnancy risk, it emerged that older women have a more positive perception than younger women and women’s concern about their child’s health is connected to the mother’s age. Older mothers are more worried and consider the risk of dying during pregnancy, preterm birth, a cesarean birth, having a newborn with a birth defect, or having one needing admission to a neonatal intensive care unit more compared to younger women [23,28,32,35,41].

To clarify inconsistencies, it is important to assume a more complex perspective on the issue. For instance, in another study [24], the authors verified that prenatal depressive symptoms were higher in the group of older women during the prenatal period, whereas depressive symptoms and parenting stress in the postpartum period increased at all ages. Moreover, cortisol levels increased starting from 30 years of age but only at 3 months after birth. In contrast, family functioning, social support, and maternal attitudes improved starting from 30 years of age, underlining that some benefits may compensate for other disadvantages. Consistently, other studies reported that despite the higher depressive symptoms during the prenatal period in pregnant women aged over 35, social support (especially objective support and subjective support), family functioning, and maternal attitudes improved with age and had stronger protective effects against depression for women of advanced maternal age compared to women aged under 35 [20,60].

In the case of contradictory results, the different methods used in the studies about the selection of the participants, the analyses, the instruments administered, and specific objects should also be considered.

### 3.6. Parenting Behaviors of Advanced Age Mothers

Regarding the parenting behaviors of advanced-age mothers, some studies suggest that increasing maternal age is linked to a more effective parenting style. Specifically, older mothers seem to be more responsive, supportive, closer, and less strict with their children [21,34]. In turn, as maternal age grows, children seem to show fewer behavioral, social, and emotional problems. These data have been confirmed longitudinally from 3 to 15 years of age [20,21,22]. 

## 4. Discussion

An increasing number of women in developed nations are choosing to postpone childbirth until their late thirties and beyond. These women express varying degrees of support from healthcare professionals.

The parenting literature focused on older mothers has typically investigated the higher obstetrical complications in older women compared to younger women. Low birth weight, preterm or post-term birth, postpartum hemorrhages, placental detachment, placenta previa, and spontaneous abortion have been proven to be among the more common adverse outcomes [8,9,10,11,12,14,15,16,17,19]. 

However, there is a paucity of work in the literature on the psychological implications of maternity at an advanced age for both mothers and their children. 

The present scoping review has pointed out that there are some positive effects of becoming mothers at older ages. Indeed, older mothers tend to have higher socio-economic levels and feel more secure about their working position. Usually, they are also in a stable relationship. These conditions allow woman to dedicate more resources to their children [29], playing a fundamental role in favoring their well-being. 

Moreover, several studies have highlighted how older mothers are likely to pay more attention to the quality of their relationship with their offspring. Certainly, this is possible because these mothers feel more content as parents and are better prepared to take care of their infants. Consequently, their parenting style is characterized by supportiveness and closeness to their child. These dimensions promote a child’s positive physical and mental well-being [25,34]. In fact, children born to older women have fewer behavioral, social, and emotional difficulties, as well as better language ability [29,34,62], supporting health throughout development up to adulthood. However, this outcome is strongly linked to several concurrent compensating variables, including, above all, the low psychosocial risk background of the studied AMAs. In addition, numerous women indicate their dependence on friends for support, particularly through the exchange of experiences. A significant number of women highlight the advantage of being of advanced maternal age, as others within their circle of friends have already experienced childbirth and could offer valuable insights and information [63]. 

To summarize, in contrast to the proven physical health risks for older age mothers and their offspring, it seems that advanced maternal age may play a protective influence on behavioral, social, and emotional difficulties for children.

A change in dialogue and ongoing scientific findings regarding certain risks associated with advanced paternal age entering clinical application could transfer the responsibility from predominantly resting on women to being shared by both men and women [64].

In conclusion, this review indicates the need to consider advanced maternal age in a more complex and multifactorial perspective.

Healthcare practitioners can play a crucial role in cultivating trusting relationships and empowering interactions with women. They can offer a range of resources for childbirth preparation and provide sensitive assistance in birth planning, aiming to support the agency of women while effectively managing potential risks. Implementing guidelines and protocols in the antenatal care of older pregnant women based on empirical evidence could promote consistency, especially in terms of risk management and the childbirth process.

### 4.1. Studies Limits

This review presents different limitations. First, the different methodological approaches, as well as different sample sizes, used in studies included in this review make it challenging to identify the specific impact of advanced maternal age on the considered outcomes. In line with this, not all studies consider the several protective factors that older mothers may benefit from. 

Also, different studies did not focus on the same outcome variables of advanced maternal age; some focused on women’s mental health, others on child development outcomes, and the mother–infant relationship.

Furthermore, most studies were carried out without considering the impact of the maternal educational level and economic status; however, when this was performed, these variables represented a protective factor for the outcomes, regardless of the mother’s age.

In addition, most studies were conducted with samples of women from Western countries in which the mean maternal age at childbirth is higher compared to low-resourced countries.

Finally, as is typically conducted in scoping reviews, no systematic quality assessment of the included publications was performed. 

For all the above reasons, interpreting this review’s conclusions must be undertaken with caution.

### 4.2. Future Directions

More research is still needed; future studies should differentiate between desired and unwanted pregnancies; finally, outcomes must take into consideration the impact of medically assisted pregnancies and acknowledge the significant role of fathers.

Therefore, it is necessary to follow women during the perinatal period, identifying and supporting protective factors. The maturity and financial stability reached with age surely allow these mothers to dedicate more attention to their children. The physical presence and empathic and caring relationship that mothers have with their infants have long-term consequences for their child’s well-being. 

Finally, as technological progress enables women to delay motherhood well into their forties, fifties, and even sixties, the term ‘very advanced maternal age’ is becoming increasingly prevalent. Subsequent research could explore the distinctive or shared experiences and needs of this older cohort of women.

In conclusion, the experience of becoming a mother in old age may be felt as desirable if women are informed of all the risks and the benefits related to age and receive, if needed, adequate support both from their social network and healthcare providers. 

## Figures and Tables

**Figure 1 behavsci-14-00147-f001:**
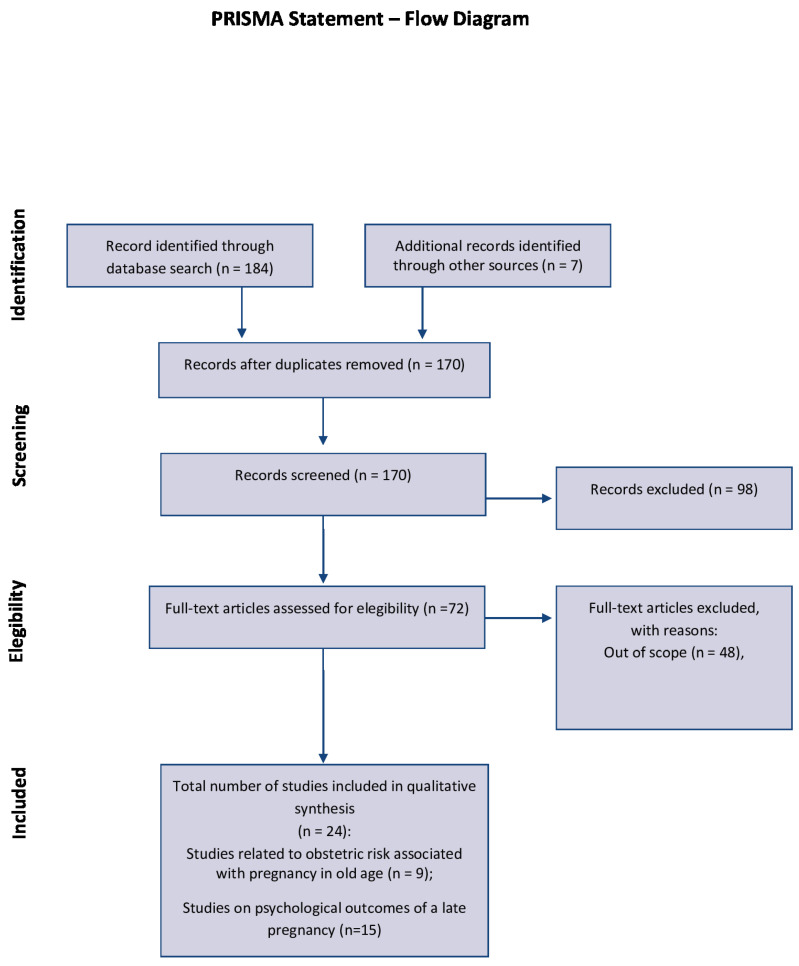
PRISMA flow diagram.

**Table 1 behavsci-14-00147-t001:** Table of included studies related to obstetric risk associated with pregnancy in old age.

Title	Authors	Sample	Research Design	Objective	Results
Pregnancy outcome in primiparae of advanced maternal age	Delbaere, I. et al., 2007 [7]	*n* = 2970 women aged ≥35 *n* = 23,921 primiparous women aged 25–29 years old	Population-based retrospective cohort study	To investigate the impact of maternal age on singleton pregnancy outcomes	Older maternal age correlated with very preterm birth, low birth weight, and perinatal death. This results independently in confounding and intermediate factors
The effect of advanced maternal age on perinatal outcomes in nulliparous singleton pregnancies	Kahveci, B. et al., 2018 [12]	*n* = 471 women <35 years *n* = 399 women 35–39 years *n* = 87 women ≥40 years	Retrospective analyses	Investigate the impact of AMA on perinatal and neonatal outcomes of nulliparous singleton pregnancies	Pregnancy at AMA is significantly associated with gestational diabetes and hypertension, preeclampsia increased cesarean section rates, SGA and spontaneous late preterm delivery
Increased maternal age, and the risk of fetal death	Fretts, R.C. et al., 1995 [8]	*n* = 94,346	Retrospective cohort study	Evaluate risk factors for fetal death among all deliveries	Women 35 years of age or older have a significantly higher rate of fetal death than their younger counterparts
Advanced maternal age and adverse pregnancy outcomes: evidence from a large contemporary cohort	Kenny, L.C. et al., 2013 [13]	*n* = 215,344 births; *n* = 122,307 mothers aged 20–29, 62,371 mothers aged 30–34, 33,966 mothers aged 35–39 7066 mothers aged ≥40	Population-based cohort study	Investigate the association between AMA (≥35) and adverse pregnancy outcomes	AMA is associated with a range of adverse pregnancy outcomes
Advanced maternal age and pregnancy outcomes: a multicountry assessment	Laopaiboon, M. et al., 2014 [14]	*n* = 308,149 singleton pregnant women	Secondary analysis of the facility-based, cross-sectional data of the WHO Multicountry Survey on Maternal and Newborn Health	Assess the association between advanced maternal age and adverse pregnancy outcomes	AMA significantly increased the risk of maternal adverse outcomes, including maternal near miss, maternal death, severe maternal outcomes, risk of stillbirths, and perinatal mortalities
Maternal risk factors for post-term pregnancy and cesarean delivery following labor induction	Roos, N. et al., 2010 [15]	*n* = 1,176,131 singletons births from gestational week 37 and onwards	Population-based cohort study	Investigate risk of post-term pregnancy (delivery at > or =42 weeks) and cesarean delivery following labor induction	Post-term pregnancy increased with increasing maternal age and was higher among primiparous women. The risk of a cesarean section following labor induction post-term, increased with maternal age and BMI: it was more than double among women ≥35
Pregnancy at or beyond the age of 40 years is associated with an increased risk of fetal death and other adverse outcomes	Hoffman, M.C. et al., 2007 [10]	*n* = 126,402 singleton deliveries divided into age groups of younger than 35 years, 35–39 years, and 40 years old or older	Retrospective study of all singleton pregnancies	Determine the frequency of fetal death in women aged ≥40 years	Pregnancy at advanced maternal age is associated with an increased rate of fetal death and other adverse pregnancy outcomes
Pregnancy outcomes at extremely advanced maternal age	Yogev, Y. et al., 2010 [17]	*n* = 177 women aged ≥45 years. Subgroup analysis compared women aged 45–49 years with women aged ≥50 years	Comparative analysis	Evaluate pregnancy outcomes in women at extremely advanced maternal age (>45 years)	Pregnancy at extreme advanced maternal age is associated with increased maternal and fetal risk
Advanced maternal age and adverse perinatal outcomes in an Asian population	Hsieh, T et al., 2010 [11]	*n* = 39,763 women, divided into age groups: Age 20–34 (*n* = 33,881), Age 35–39 (*n* = 5161), Age ≥ 40 (*n* = 721)	Retrospective cohort study	Investigate the association between AMA and adverse perinatal outcomes in an Asian population.	Advanced maternal age is associated with pregnancy complications and adverse perinatal outcomes.

**Table 2 behavsci-14-00147-t002:** Table of included studies related to psychological outcomes of a late pregnancy.

Title	Authors	Sample	Research Design	Objective	Results
Associations between advanced maternal age and psychological distress in primiparous women from early pregnancy to 18 months postpartum	Aasheim, V. et al., 2012 [20]	*n* = 19,291 nulliparous women	National cohort study	Investigate if advanced maternal age at first birth increases the risk of psychological distress during pregnancy at 17 and 30 weeks of gestation and at 6 and 18 months after birth	Women of advanced age have slightly higher scores of psychological distress during pregnancy and the first 18 months of motherhood
The parenting of preschool children by older mothers in the United Kingdom	Barnes, J. et al., 2013 [21]	*n* = 24,610	The study makes use of data from two longitudinal studies	Investigate if maternal age is relevant to parenting behavior	Punitive strategies and conflicts with children are greater in younger mothers and tend to decrease as maternal age increases. Older mothers are more supportive and their closeness to their children is greater
Comparison of perception of pregnancy risk for nulliparous women of advanced maternal age and younger age	Bayrampour, H. et al., 2012 [22]	*n* = 159 nulliparous pregnant women (105 aged 20–29 years; 54 aged 35 years or older)	Comparative descriptive study	Compare risk perception in pregnant women ≥35 with that of younger women	Women in AMA had higher medical risk scores than younger women and perceived a higher pregnancy risk for both themselves and their fetuses than younger women, including those with low-risk pregnancies
Advanced maternal age and risk perception: a qualitative study	Bayrampour, H. et al., 2012 [23]	*n* = 15 primigravidae aged 35 and above	Qualitative/ descriptive	Analyze the perception of risk for pregnant women at an AMA	Pregnancy at age ≥35 within a healthy context was perceived as a low-risk pregnancy. In the presence of other risk factors, the risk associated with age was highlighted, and women were inclined to recognize their age as a risk factor
A preliminary study to assess the impact of maternal age on stress-related variables in healthy nulliparous women	García-Blanco, A. et al., 2017 [24]	*n* = 148 nulliparus pregnant women between 18 and 40 years old	Prospective cohort study	Assess the impact of maternal age on depression, parenting stress and social functioning	Depressive symptoms showed an increase starting from 35 years old at 38 weeks of gestation, and U-shaped relationship with a minimum age of around 30 years old and 3 months after birth. Social functioning improved moderately with age
Association of maternal age with child health: a Japanese longitudinal study	Kato, T. et al., 2017 [25]	*n* = 2001 = 47,715 babies *n* = 2010 = 38,554 babies	Longitudinal study	Estimate risks of unintentional injuries and hospital admissions at 18 and 66 months, according to maternal age	The risks of unintentional injuries and hospital admissions decreased in accordance with older maternal age in both 2001 and 2010 cohorts
Age at first birth, mode of conception and psychological well-being in pregnancy: findings from the parental age and transition to parenthood Australia (PATPA) study	McMahon C.A., et al., 2011 [26]	*n* = 297 women conceiving through ART assisted reproductive technology *n* = 295 across three age groups: younger, ≤20–30 years; middle, 31–36 years; older, ≥37 years	Prospective study	Examine relationships for maternal age at first birth, mode of conception and psychosocial well-being in pregnancy	Older maternal age was associated with lower depression and anxiety symptoms, lower maternal–fetal attachment, greater resilience, and lower ratings of control in the partner relationship at a univariate level. ART, but not older maternal age, was associated with greater P–F anxiety. Older women are more resilient and report that their partners are less controlling
Older maternal age and major depressive episodes in the first two years after birth: findings from the Parental Age and Transition to Parenthood Australia (PATPA) study	McMahon, C.A., et al., 2015 [27]	*n* = 592 women in the third trimester of pregnancy in three age groups (≤30 years; 31–36 years, ≥37 years); 434 (73%) completed all assessments at four months and two years after birth	Prospective study	To ascertain if clinically significant episodes of depression between four months and two years postpartum was more common among older first-time mothers; to examine risk factors linked to the episodes of depression with a later onset within the first year after giving birth, utilizing Belsky’s model for parenting determinants	Maternal age was not related to the prevalence or timing of major depression episodes. Depression symptoms, poor child health, low practical support at four months and non-English language background predicted episodes of depression between four months and two years
First-time mothers’ pregnancy and birth experiences vary by age	Zasloff, E. et al., 2007 [28]	*n* = 1302 primiparous women	Longitudinal cohort study	Provide a comprehensive picture of the young to the old first-time mother as she presents to the clinician in terms of background, expectations, experiences and outcome of labor	Young women aged 15–20 years had the most negative expectations of the upcoming birth. The oldest women (35–43 years) did not have negative feelings about the upcoming birth during pregnancy and did not remember being afraid
Associations between older maternal age, use of sanctions, and children’s socio-emotional development at 7, 11, and 15 years	Trillingsgaard, T. et al., 2018 [29]	*n* = 4741 mothers		Analyze if older maternal age is associated with less sanctions and with positive child outcome at ages 7, 11, and 15	Older maternal age was associated with less frequent use of verbal and physical sanctions towards children at ages 7 and 11. At age 15, the association remained significant for verbal but not physical sanctions. Older maternal age was associated with fewer behavioral, social, and emotional difficulties in children at age 7 and age 11 but not at age 15
Advantages of later motherhood	Myrskylä, M. et al., 2013 [30]	Review	Comprehensive review of the existing literature	Summarize the literature on the benefits of advanced maternal age	Children who are born to older mothers tend to have higher cognitive scores than those with younger parents. Older maternal age is often associated with socioeconomic resources that may help to alleviate the stress of caring for a child
“Doing it properly”: the experience of first mothering over 35 years	Carolan, M. 2005 [31]	*n* = 22 primiparous women aged from 35 to 48 years over three junctures: at 35–38 weeks gestation, 10–14 days postpartum, and 8 months postpartum	Longitudinal, qualitative study	Exploring participants ‘ experiences of childbirth and early parenting	Women worry about risks associated with age and about the lack of information about it. They have difficulties adjusting throughout the postpartum period
Late motherhood: the experience of parturition for first-time mothers aged over 35 years	Carolan, M 2003 [32].	*n* = 20 primiparae aged >35 over three junctures: 35–38 weeks gestation; 7–10 days postpartum; and 6–8 months postpartum	Longitudinal, qualitative study	To explore the participants’ experience of birth and early parenting	Women feel more prepared and more responsible. However, they expressed concern about their career and the need for more information during perinatality. They also felt discriminated because of their late pregnancy
The project: having a baby over 35 years	Carolan, M. 2007 [33]	*n* = 22 women aged >35	Longitudinal, qualitative study	To evaluate the experiences of a group of first-time mothers aged more than 35 years.	The results showed that some mums viewed having children as a significant life project. Data analysis showed that the project moved through distinct phases, including gathering information, planning, and completing chores before the birth (cleaning the deck)
The health and development of children born to older mothers in the United Kingdom: observational study using longitudinal cohort data	Sutcliffe, A.G. et al., 2012 [34]	*n* = 31,257 children aged 9 months *n* = 24,781 children aged 3 years *n* = 22,504 children age 5 years.	Observational study of longitudinal cohorts	Assess relationships between children’s health, development and maternal age	The risk of children having unintentional injuries requiring medical attention or hospital both decreased as maternal age increased. Language development, fewer social and emotional difficulties were associated with improvements in increasing maternal age. The children of teenage mothers had more difficulties than children of mothers aged 40.

## Data Availability

Data sharing is not applicable to this article.

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
