# Peer review of "Advanced Maternal Age: A Scoping Review about the Psychological Impact on Mothers, Infants, and Their Relationship"

_behavsci, 2024, doi:10.3390/bs14030147_

Round 1
Reviewer 1 Report
Comments and Suggestions for Authors
This manuscript is a narrative review of the literature on the psychological effects on mothers, infants, and their relationship.
Although the topic is interesting, I find there are a lot of issues with the manuscript. First of all the authors use different names for the review they conducted including narrative review, scoping review and rapid evidence review. These are not the same and it is not clear to me what type of review they have conducted.
The introduction is quite short and does not really address why this study is important. The authors provide a brief overview of ‘problems’ or issues related to older maternal age but they note nothing on studies focusing on benefits of maternal age as these studies have been done.
Under search strategy, the authors mention that the databases were searched from their inception until 2023. I am confused why such a wide range and I assume not all of these databases had then same inception dates? It is important to include this information. As the authors mention that the average maternal age in Europe has increased, why would you focus on studies that were done a long time ago as this phenomena was not as prevalent several decades ago and might not be applicable to the current situation?
Under population, the authors state that women under 35 were considered, however, I confused about the reasoning? Are they considered when compared with younger women? If so, how are the younger women defined then?
How is psychological effect defined? The authors state that they want to compare older mothers’ experiences with younger mothers. However, if you include qualitative data, you might not be able to examine this.
Study design: are the studies quantitative or qualitative or mixed methods? Please specify.
In the data extraction, the authors mention that they included studies that focused on the psychological outcome (which aligns with the title) but also obstetric risk, which is not mentioned in the title. This outcome was also not defined in the inclusion criteria.
Looking at the table, I am not sure why Kato et al. study is included as I thought the focus was on psychological effects and this study focuses on unintentional injuries and hospital admissions?
Page 9, line 131: mothers in advanced maternal age had higher perception of pregnancy risk compared to younger women but only when other risk factors were included. Did this not hold for younger women, so did younger women experienced fewer risks even when the risk factors were included?
After reading the results, it is still not clear to me whether this is a traditional or narrative review or a scoping review. It is not clear to me how the articles were coded or analysed. Do the authors take into consideration the sample sizes of each of these studies? Some studies might have small sample sizes and then become difficult to compare with studies with large sample sizes.
I would restructure the results section and first focus on countries, participants etc. and then have the actual results at the end.
How many studies exactly were there from each of the mentioned countries? This is important to know as Western countries tend to have higher maternal age at first pregnancy which influences the results.
I would suggest putting all the measures in a table, instead of trying to include them in the text.
3.5 not only mothers
Why do the authors only mention that in the past 40 years in the United States more infants have older fathers. This is not only the case in the U.S. but in many other Western countries as well.
I think it is important to acknowledge fathers but I would be hesitant to base the findings on for example IQ tests or including the child’s gender. The authors seem to stray away from the aims of the study by including these details.
Page 12, line 241. This information should be included in the introduction, not just the discussion as it was not clear why these studies focusing on physical risks were included.
As the authors mentioned at the beginning of the article, many of the positive benefits of an older mother are related to maternal education and age. It is important to acknowledge this clearly and that samples might be skewed to include primarily those mothers, while this might not be the case for mothers who are from CALD communities or lower SES or education.
There is no discussion about who these mothers were in all these studies? It sounds like most of them were from high SES families? What about diversity in ethnicity and SES?
Page 12, line 279, what kind of guidelines are the authors referring to?
Line 291, the authors mention that this is a rapid evidence review so I am totally confused about what kind of review the authors conducted as there needs to be consistency in the wording.
Line 296, what about pregnancy loss?
The authors should also acknowledge the higher SES and education levels that often come along with delayed motherhood.
Page 1, line 32, a financial stability should be financial stability
Page 1, line 36, what does ‘such condition’ refer to? I would suggest saying something like ‘the increased maternal age carries with it….
Section 3, first sentence ‘as regards’ should be ‘in regards to’
Page 10, line 40, replace ‘proved’ with showed.
Page 10, line 158. I think the authors mean ‘contradictory’ instead of ‘controversial’
Page 11, line 186: datum should be data.
Page 11, line 204, as regards should be in regards.
Page 11, line 228, ‘of’ should be ‘as’.
Comments on the Quality of English Language
Will need a review as there are some grammatical and spelling errors.
Reviewer 2 Report
Comments and Suggestions for Authors
This is a valuable paper. However, it would benefit from some clarification.
Here are a list of points to consider:
Line 21: “appears to exert a protective influence on child’s behavioral” change to “appears to exert a protective influence on children’s behavioral”
Lines 29 to 30: “In recent years, the age of first-time mothers has significantly increased compared to the past.” suggest delete “compared to the past”
Line 32: “to reach a financial stability” change to “to reach financial stability”
Lines 32 to 33: “may lead to postpone the moment in which women decide to become mothers” change to “may lead women to postpone the moment in which they decide to become mothers”
Line 34: “29,7 years” suggest change to “29.7 years”
Line 36: “Such condition” change to “These conditions”
Line 57: “the consultation with the stakeholders” change to “consultation with the stakeholders”
Lines 61 to 63: “advanced maternal age” appears twice in the list of keywords
Line 70: “aim to comparing pregnancy outcomes” change to “aim of comparing pregnancy outcomes”
Line 91: “including criteria” change to “inclusion criteria”
Lines 138 to 139: “Maternal awareness of risk is a key element that can also have a positive influence on the offspring.” this statement should be clarified. What is the positive influence on the offspring?
Line 158: “Instead, results are controversial in relation to the presence of psychological distress” change to “By contrast, results are mixed in relation to the presence of psychological distress”
Line 164: “and a slightly higher” change to “and slightly higher”
Lines 165 to 166: “emerged that the older women have a better perception than younger” change to “it emerged that the older women have a more positive perception than younger women”
Lines 166 to 167: “women’s concern about the child’s health is connected to mother’s age” clarify which direction the association is in: are older mothers more concerned about their children’s health or less concerned?
Line 171: “in the postpartum” change to “in the postpartum period”
Lines 189 to 220: I think these lines should go at the beginning of the Results section
Lines 221 to 235: I suggest that these results concerning fathers would be better in a separate paper focussing on older fathers.
Lines 237 to 266: Some of the material in the discussion section reads more like results. Any results mentioned for the first time should be in the results section, with the Discussion sections kept strictly for discussion.
Lines 459 to 470: There is a formatting issue towards the end of the References section, with the reference numbers being repeated at the beginning of each reference.
Comments on the Quality of English Language
I have made a number of suggestions for clarifications to the English in the "Comments and Suggestions for Authors
" section.
Reviewer 3 Report
Comments and Suggestions for Authors
1. Several studies reported contradictory findings regarding the influence of AMA on mothers' and children's psychological adjustment. It would be great if the authors could discuss potential reasons why the contradictory findings emerge.
2. Effects of maternal age on mothers' and children's psychological adjustment may not be linear. Very low maternal age may also be risky. After excluding very young mothers, do mothers above 35 years still have generally better psychological adjustment?
3. When examining the effects of AMA, did these studies controlled for maternal education and other indicators of social-economic status?
